# BALANCING THE PICTURE: DEBIASING VISION-LANGUAGE DATASETS WITH SYNTHETIC CONTRAST SETS

## ABSTRACT

Vision-language models are growing in popularity and public visibility to generate, edit, and caption images at scale; but their outputs can perpetuate and amplify societal biases learned during pre-training on uncurated image-text pairs from the internet. Although debiasing methods have been proposed, we argue that these measurements of *model bias* lack validity due to *dataset bias*. We demonstrate there are spurious correlations in COCO Captions, the most commonly used dataset for evaluating bias, between background context and the gender of people in-situ. This is problematic because commonly-used bias metrics (such as Bias@K) rely on per-gender base rates. To address this issue, we propose a novel dataset debiasing pipeline to augment the COCO dataset with synthetic, gender-balanced contrast sets, where only the gender of the subject is edited and the background is fixed. As existing image editing methods have limitations and sometimes produce low-quality images; we introduce a method to automatically filter the generated images based on their similarity to real images. Using our balanced synthetic contrast sets, we benchmark bias in multiple CLIP-based models, demonstrating how metrics are skewed by imbalance in the original COCO images. Our results indicate that the proposed approach improves the validity of the evaluation, ultimately contributing to more realistic understanding of bias in CLIP.

## 1 INTRODUCTION

Vision-Language Models (VLMs) are rapidly advancing in capability and have witnessed a dramatic growth in public visibility: DALL-E Ramesh et al. (2021) has more than 1.5 million users creating over 2 million images a day; the discord channel for MidJourney MidJourney (2023) hosts over two million members Salkowitz (2022); and shortly after its release, Stability.AI reported that their Stable Diffusion model Rombach et al. (2022) had over 10 million daily active users Fatunde & Tse (2022). Underpinning these powerful generative models are image-text encoders like CLIP Radford et al. (2021), which are themselves used for many discriminative tasks, such as video action recognition, open set detection and segmentation, and captioning. These encoders are pre-trained on large-scale internet scraped datasets. The uncurated nature of such datasets can translate to generated images that risk inflicting a range of downstream harms on their end users and society at large – from bias and negative stereotypes, to nudity and sexual content, or violent or graphic imagery Birhane et al. (2021); Cherti et al. (2022).

In light of these issues, coupled with growing use of generative AI, it is vital to reliably benchmark the bias in VLMs, particularly in the image-text encoders. A small emerging body of work attempts to measure bias in VLMs Agarwal et al. (2021); Berg et al. (2022); Chuang et al. (2023), or to debias their feature representations Berg et al. (2022); Chuang et al. (2023). Yet the legitimacy of this work critically depends on both a suitable evaluation metric and an evaluation dataset to accurately depict the bias in pre-trained model weights and reliably signal whether debiasing attempts have been successful. The predominant focus on model-centric debiasing methods has overshadowed two main challenges associated with datasets and metrics: (i) the common use of cropped face datasets, such as FairFace Kärkkäinen & Joo (2021), fall short because excluding contextual background presents an inaccurate and unreliable assessment of bias in natural images; and (ii) even if natural, open-domain images containing contextual clues are used, they are unbalanced by identity attribute representation

within contexts. This is problematic because commonly-used bias metrics, such as Bias@K, are affected by the naturally-occurring distribution of images. Thus, while using contextual images is desirable, it comes at the cost of spurious correlations, affecting the reliability of bias metrics.

In this paper, we argue that these confounding factors arising from the interaction of metric choice and biased datasets paint an unreliable picture when measuring model bias in VLMs. To counter these issues, we propose a synthetic pipeline for debiasing a dataset into contrast sets balanced by identity attributes across background contexts. Our pipeline draws on the success of contrast sets in NLPs Gardner et al. (2020) and leverages recent advances in controllable image editing and generation Brooks et al. (2022). We illustrate our approach with a focus on gender bias and define a contrast set as containing pairs of images from COCO Chen et al. (2015) where each image ID has two synthetically-edited versions (one man, one woman) where the background is fixed and only the person bounding box is edited. We make three key contributions: (1) We demonstrate spurious correlations in the COCO dataset between gender and context, and show their problematic effects when used to measure model bias (Sec. 3); (2) We present the GENSYNTH dataset, built from a generative pipeline for synthetic image editing, and a filtering pipeline using KNN with real and synthetic images to control for the quality of the generated images (Sec. 4); (3) We benchmark CLIP models Radford et al. (2021); Wang et al. (2021b) on our GENSYNTH dataset, which has no spurious correlation, and cast doubts on the effectiveness of debiasing methods (Sec. 5).

Our findings demonstrate that debiasing a dataset with synthetic contrast sets can avoid spurious correlations and more reliably measure model bias. While synthetically-edited data has promise in (i) preserving privacy of subjects included in vision datasets, and (ii) adding controllability to the dataset features, it also risks introducing a real-synthetic distribution shift and stacking biases of various generative models may essentialise representations of gender (see Sec. 6). Despite these early-stage limitations, this work starts a conversation about the importance of the interaction between dataset features with bias metrics, ultimately contributing to future work that paints a more accurate and balanced picture of identity-based bias in VLMs.

## 2 RELATED WORKS

**Defining Fairness and Bias.** Fairness is a complex, context-dependent concept Mehrabi et al. (2021); Verma & Rubin (2018). Here, we adopt a narrow definition where no group is advantaged or disadvantaged based on the protected attribute of gender in retrieval settings Friedrich et al. (2023); Hendricks et al. (2018). The metrics employed in this paper, *Bias@K* Wang et al. (2021a) and *Skew@K*, Geyik et al. (2019) are used to assess disparity in distribution between search results and desired outcomes. In this work, we assume activities such as *dancing, skateboarding, laughing* would not have a strong gendered prior and thus the desired distribution is one where all protected attributes have equal chance of being returned in a query that does not explicitly mention gender.[1]

**Measuring Model Bias.** Measuring bias in VLMs is a growing area of research Luccioni et al. (2023). Early work measures the misclassification rates of faces into harmful categories Agarwal et al. (2021). Several works measure outcome bias for text-to-face retrieval Berg et al. (2022); Chuang et al. (2023); Seth et al. (2023), though it is unclear how such measurements made on cropped face datasets generalise to real-world settings. For gender fairness in open-domain images, COCO Captions Chen et al. (2015) is a standard benchmark for cross-modal retrieval Wang et al. (2021a; 2022) and image captioning Hendricks et al. (2018); Zhao et al. (2021).

**Dataset Bias.** Datasets, including those used for bias evaluation, have their own biases from curation and annotation artefacts. Image datasets have been found to include imbalanced demographic representation Buolamwini & Gebru (2018); De Vries et al. (2019); Torralba & Efros (2011); Wang & Russakovsky (2023); Wang et al. (2019); Zhao et al. (2021), stereotypical portrayals Caliskan et al. (2017); Schwemmer et al. (2020); van Miltenburg (2016), or graphic, sexually-explicit and other harmful content Birhane et al. (2021). Similar to Meister et al. (2022); Wang & Russakovsky (2023), we identify spurious gender correlations in the COCO Captions dataset and show this renders the datasets unsuitable for current bias retrieval metrics. Techniques to reduce dataset biases range

---

[1]In certain specific contexts, for example, pregnant or breastfeeding women, we may not necessarily want an equal distribution of masculine and feminine images to be returned, though we must be careful to not conflate biological gender and gender identity (see Sec. 6).

from automatic Schuhmann et al. (2022) to manual filtering Yang et al. (2020) of harmful images, such as those containing nudity Schuhmann et al. (2022), toxicity, or personal and identifiable information Asano et al. (2021). Yet, these filters cannot identify subtle stereotypes and spurious correlations present in open-domain images – making it difficult to curate a wholly unbiased natural image dataset Meister et al. (2022).

**Mitigating Dataset Bias with Synthetic Data.** Deep networks need large amounts of labeled data, prompting the creation of synthetic datasets for various computer vision tasks Dosovitskiy et al. (2015); Johnson et al. (2017); Michieli et al. (2020); Song et al. (2017). More recently, progress in generative models Ramesh et al. (2021); Rombach et al. (2022); Saharia et al. (2022) has enabled methods to synthetically generate training data Brooks et al. (2022); Li et al. (2022); Peebles et al. (2022); Zhai & Wu (2018). Similarly, text-guided editing methods Brooks et al. (2022); Hertz et al. (2022); Tumanyan et al. (2022) offer scalable and controllable image editing, potentially enhancing dataset fairness and removing issues related to existing spurious correlations. Several works propose the use of synthetic datasets for mitigating dataset bias, such as with GANs Sattigeri et al. (2019) or diffusion models Friedrich et al. (2023). However, synthetic or generated data may not necessarily represent underlying distributions of marginalised groups within populations and thus still unfairly disadvantage certain groups Altman (2021); Belgodere et al. (2023); Bhanot et al. (2021); Lu et al. (2023). To combat these risks, fairness in generative models is an area gaining popularity: StyleGan Karras et al. (2019) has been used to edit images on a spectrum, rather than using binary categories Hermes (2022); Friedrich et al. (2023) use human feedback to guide diffusion models to generate diverse human images; and Kim et al. (2022) learn to transfer age, race and gender across images. Similar to our work, GAN-based frameworks Denton et al. (2019); Ramaswamy et al. (2021) edit an *existing* face dataset to equalise attributes and enforce fairness. Our work extends this approach to open-domain images, introducing an automatic filtering technique for improving the quality of edits. To our knowledge, we are the first to propose image editing of open-domain images for fairness. Our work is also inspired by the use of contrast sets in NLP Gardner et al. (2020), which have been used to alter data by perturbing demographics (race, age, gender) in order to improve fairness Qian et al. (2022). We use synthetically-generated contrast sets by augmenting both the textual and visual input to CLIP, for a more accurate evaluation of VLM bias.

## 3 MEASURING GENDER BIAS ON NATURAL IMAGES

While prior works make in-depth comparisons between models, and even metrics Berg et al. (2022), there is a dearth of research investigating whether natural image datasets, with their own biased and spurious correlations, are suitable benchmarks to measure bias in VLMs. In this section, we investigate the extent of dataset bias from spurious correlations in COCO (Sec. 3.3) and its effect on reliably measuring model bias (Sec. 3.4).

### 3.1 PRELIMINARIES

We first define the bias metrics and the framework used to measure model bias on image-caption data.

**Bias@K** Wang et al. (2021a) measures the proportions of masculine and feminine images in the retrievals of a search result with a gender-neutral text query. For an image $I$, we define a function $g(I) = $ male if there are only individuals who appear as men in the image, and $g(I) = $ female if there are only individuals who appear as women. Given a set of $K$ retrieved images $\mathcal{R}_K(q)$ for a query $q$, we count the images of apparent men and women as:

$$N_{\text{male}} = \sum_{I \in \mathcal{R}_K(q)} \mathbb{1}[g(I) = \text{male}] \quad \text{and} \quad N_{\text{female}} = \sum_{I \in \mathcal{R}_K(q)} \mathbb{1}[g(I) = \text{female}].$$

We define the gender bias metric as:

$$\delta_K(q) = \begin{cases} 0, & N_{\text{male}} + N_{\text{female}} = 0 \\ \frac{N_{\text{male}} - N_{\text{female}}}{N_{\text{male}} + N_{\text{female}}}, & \text{otherwise.} \end{cases}$$

For a whole query set $Q$, we define:

$$\text{Bias@K} = \frac{1}{|Q|} \sum_{q \in Q} \delta_K(q). \tag{1}$$

**Skew@K** Berg et al. (2022); Geyik et al. (2019) measures the difference between the desired proportion of image attributes in $\mathcal{R}_k(q)$ for the query $q$ and the actual proportion. Let the desired proportion of images with attribute label $A$ in the set of retrieved images be $p_{d,q,A} \in [0,1]$ and the actual proportion be $p_{\mathcal{R}(q),q,A} \in [0,1]$. The Skew@K of $\mathcal{R}(q)$ for an attribute label $A \in \mathcal{A}$ is:

$$\text{Skew@K}(\mathcal{R}(q)) = \ln \frac{p_{\mathcal{R}_K(q),q,A}}{p_{d,q,A}}, \tag{2}$$

where the desired proportion $p_{d,q,A}$ is the actual attribute distribution over the entire dataset. A disadvantage of Skew@K is that it only measures bias with respect to a single attribute at a time and must be aggregated to give a holistic view of the bias over all attributes. We follow Berg et al. (2022) and take the maximum Skew@K among all attribute labels $A$ of the images for a given text query $q$:

$$\text{MaxSkew@K}(\mathcal{R}(q)) = \max_{A_i \in \mathcal{A}} \text{Skew}_{A_i}\text{@K}(\mathcal{R}(q)), \tag{3}$$

which gives us the "largest unfair advantage" Geyik et al. (2019) belonging to images within a given attribute. In our work, a MaxSkew@K of 0 for the attribute gender and a given text query $q$ implies that men and women are equally represented in the retrieved set of $K$ images $\mathcal{R}_K(q)$. We ignore all images with undefined attribute labels (in this case gender) when measuring MaxSkew@K.

**COCO** is a dataset of 118k images with detection, segmentation and caption annotations, covering 80 distinct categories, including people Chen et al. (2015); Lin et al. (2014). Each image has five captions written by different human annotators. COCO is commonly used to measure gender bias in VLMs in tandem with the Bias@K metric Chuang et al. (2023); Wang et al. (2021a; 2022).

## 3.2 Gendered Captions and Images in COCO

The bias metrics defined in Sec. 3.1 require gender attribute labels for each image and gender-neutral text queries, but these are not naturally present in captioned image data such as COCO. We describe the steps to automatically label gender for images and to neutralise gender information in captions.

**Extracting Image Gender Labels from Captions.** We assign a gender label to each COCO image, following prior work Wang et al. (2021a). For each image, we concatenate all five captions into a single paragraph. If the paragraph contains only feminine words and no masculine words, the image is assigned a female label, and vice versa. If the paragraph contains words from both or neither genders, it is labeled as undefined. The full list of gendered words is detailed in the Appendix. Using this procedure, we implement the function $g$ in Sec. 3.1. The COCO 2017 train set contains 118,287 images, of which 30,541 (25.8%) are male, 11,781 (9.9%) are female, and 75,965 (64.2%) are undefined. The COCO 2017 validation set contains 5,000 images, of which 1,275 (25.5%), are male, 539 (10.8%) female, and 3,186 (63.7%) undefined. This procedure gives high precision in the gender-pseudo label, as any ambiguous samples are rejected. However, images may be incorrectly labeled as undefined (lower recall) due to, for example, misspelling of the gendered words in the human-annotated captions or omission of rarer gendered terms in our keyword list.

**Constructing Gender-Neutral Captions.** We construct gender-neutral captions by replacing gendered words with neutral ones, e.g. "man" or "woman" become "person", and the sentence "A *man* sleeping with *his* cat next to *him*" becomes "A *person* sleeping with *their* car next to *them*". The full mapping of gender-neutral words and more examples of original and neutralised captions are in the Appendix.

## 3.3 Identifying Spurious Correlations with Gender

As reported above, COCO contains more than twice as many male images as it does female ones. This will inevitably affect retrieval-based bias metrics, as there will be more male images in the retrievals. One naïve way to fix this is to undersample the male images in order to arrive at a *Balanced* COCO dataset. However, ensuring equal distribution of demographic attributes does not necessarily ensure the dataset is unbiased as a whole. Spurious correlations can result in subsets of the data being highly correlated with certain attributes. Here we explore whether for certain contexts in the COCO dataset, e.g., skateboarding, one gender is over-represented. We take two approaches to evidence these spurious correlations.

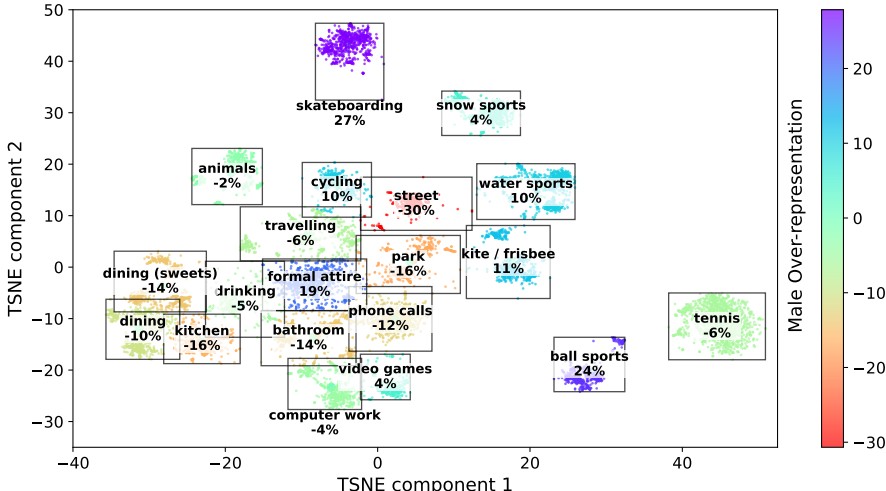

Figure 1: t-SNE clusters ($M = 20$) of gender-neutralised caption embeddings. Each cluster is manually assigned a name, then coloured and labelled according to its male over-representation factor. The male over-representation factor is the difference between the percentage of male images in the particular cluster and the percentage of male images overall in the dataset.

**K-means Clusters with Caption Embeddings.** First, we find semantic clusters of captions and evaluate the gender balance within them. For every image $I_n$, we embed its gender-neutralised captions $C_n^k$, where $k = \{1, \ldots, K\}$ represents the $K$ captions of the image, with RoBERTa Liu et al. (2019) to get features $f_n^k$. We average the features to get $f_n = \frac{1}{K} \sum_{k=1}^{K} f_n^k$. Next, we cluster the features $f_n, n = \{1, \ldots, N\}$ into $M = 20$ clusters with K-Means. Finally, for each cluster, we extract salient words using Latent Dirichlet Allocation (LDA) and give a manually-defined cluster label. In Fig. 1 we show a t-SNE representation of the discovered clusters, together with the degree of male over-representation. We see that in sports-related concepts men are over-represented, whereas in scenes in kitchens, bathrooms, streets, and parks, women are over-represented. For a comparison of this analysis to a keyword-based analysis, and a list of all discovered classes and salient words according to LDA, please refer to the Appendix.

**Spurious Correlations Classifier.** Following Schwemmer et al. (2020), we investigate the presence of spurious correlations by training classifiers to predict binary gender labels of images and captions where the explicit gender information is removed for both training and testing. Specifically, for the image classifier (ResNet-50) we replace all person bounding boxes with black pixels; and for the caption classifier (BERT-base) we use the gender-neutralised captions. The training and testing data is COCO train and validation defined in Sec. 3.2, but with undefined images dropped. On unseen data, the text-only classifier on gender-neutralised captions achieves 78.0% AUC and the image-only classifier on person-masked images achieves 63.4% AUC. Given that a random chance model achieves 50% AUC and an image classifier on unmasked images achieves 71.9% AUC, it is clear that spurious correlations in the image, as well as biases in the caption, provide a significant signal to predict gender of the person in the image even when there is no explicit gender information.

## 3.4 THE EFFECT OF DATASET BIAS ON MODEL BIAS MEASUREMENT

There are two angles to dataset bias: (1) over representation – there are more photos of men than women in COCO on average, and (2) spurious correlations, i.e., some environments/backgrounds are more prevalent for specific gender groups. Accordingly, the dataset used for bias evaluation significantly affects the model bias measurement. This is exemplified by a theoretically gender-agnostic model, which we instantiate as a TF-IDF (Term Frequency - Inverse Document Frequency) ranking model for caption-to-caption retrieval on gender-neutralised captions. Despite being based on a simple numerical statistic of word occurrences, devoid of any inherent gender bias, this model still exhibits non-zero bias when evaluated on COCO captions. Our findings, reported in Tab. 1, include Bias@K and MaxSkew@K measurements on COCO Val, compared against a random model and

Table 1: Comparison of model gender bias for CLIP Radford et al. (2021), a theoretically gender-agnostic model (TF-IDF on non-gendered words) and a random model, on the COCO validation set under unbalanced and balanced (with standard deviation computed over 5 runs) settings.

| Model | COCO Val | | | | COCO Val (Balanced) | | | |
|---|---|---|---|---|---|---|---|---|
| | Bias@K | | MaxSkew@K | | Bias@K | | MaxSkew@K | |
| | K=5 | K=10 | K=25 | K=100 | K=5 | K=10 | K=25 | K=100 |
| Random Model | 0.37 | 0.40 | 0.15 | 0.06 | $0.00_{\pm 0.07}$ | $0.00_{\pm 0.07}$ | $0.14_{\pm 0.00}$ | $0.07_{\pm 0.00}$ |
| TF-IDF$_{gender-agnostic}$ | 0.22 | 0.24 | 0.29 | 0.22 | $-0.06_{\pm 0.00}$ | $-0.08_{\pm 0.00}$ | $0.25_{\pm 0.00}$ | $0.18_{\pm 0.00}$ |
| CLIP | 0.20 | 0.23 | 0.28 | 0.23 | $-0.03_{\pm 0.01}$ | $-0.06_{\pm 0.01}$ | $0.24_{\pm 0.00}$ | $0.19_{\pm 0.01}$ |

CLIP. For Balanced COCO Val, all models register an approximate Bias@K of zero, a consequence of the metric's signed nature that tends to average towards zero over many directions of spurious correlations on biased but balanced data. Yet, for unbalanced data, Bias@K shifts towards the over-represented attribute in the dataset, making it an unsuitable metric for model bias measurement, as it reflects dataset bias instead. MaxSkew@K, despite being an absolute measure, is not exempt from these issues. It still records large values for the theoretically gender-agnostic model and the random model, suggesting that the established framework may be inadequate for bias measurement on natural image datasets that inherently possess their own biases. The experiments in Tab. 1 show the inadequacy of Bias@K and MaxSkew@K for measuring bias on natural image datasets that are imbalanced. Even when correcting for the imbalance issue, spurious correlations remain in the data. Therefore, we argue that a balanced unbiased dataset is required to robustly measure bias and compare different pre-trained models and debiasing strategies. This motivated us to propose GENSYNTH in Sec. 4, which attempts to fix both issues.

## 4 GENSYNTH: A SYNTHETIC GENDER-BALANCED DATASET USING CONTRAST SETS

Given the limitations of measuring Bias@K and MaxSkew@K on natural images and the spurious correlations in existing datasets, we propose a framework for editing natural images into *synthetic contrast sets* that remove spurious background correlations along the attribute of interest and apply the pipeline on COCO to obtain the GENSYNTH dataset (see Fig. 2). We first synthetically edit the person in images to cover both gender labels with fixed background context (Sec. 4.1), followed by automatic filtering that ensures the quality and correctness of the edited persons (Sec. 4.2). Finally, we verify the quality of the edited images and the filtering method (Sec. 4.3). While we implement this for the gender attribute, in practice, our pipeline could be used to generate synthetic contrast sets for other identity attributes, requiring only the availability of person bounding boxes for the images.

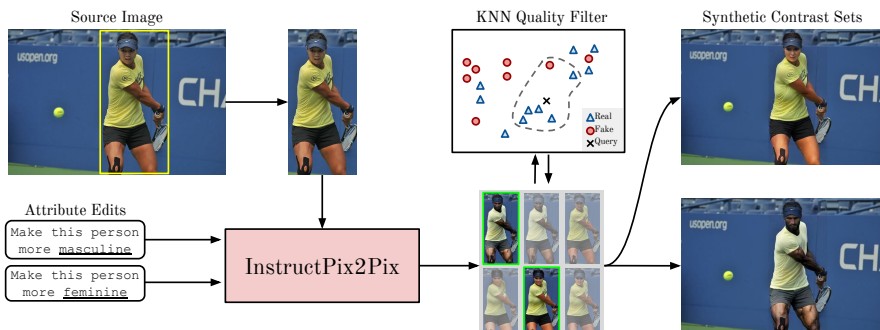

Figure 2: An overview of our pipeline for *dataset debiasing* across a target attribute, in this case gender, ensuring equal demographic representation. A source image containing a person is given as input to InstructPix2Pix along with instructions to synthesise each attribute label. The resulting edits are filtered for quality via K-Nearest Neighbour (KNN) thresholding to ensure realistic-looking edits for each attribute label (male and female).

## 4.1 SYNTHETICALLY EDITING IMAGES

Leveraging advancements in text-conditioned image generation and editing, we use an instruction-based model, InstructPix2Pix Brooks et al. (2022), for editing objects in an image – referred to as the *source* image – while keeping the background unchanged. We edit source images from COCO that (i) contain only one person, inferred from the number of person bounding boxes; and (ii) have a defined gender label, as defined in Sec. 3.2. These restrictions remove ambiguity. Next, we crop the image to the single person bounding box and feed it to InstructPix2Pix Brooks et al. (2022) along with multiple edit instructions for each attribute label, e.g., "make this person more *masculine/feminine*". Refer to Tab. 7 for the complete set of edit instruction templates. The edited person is then replaced in the source image. By only editing the appearance of the person in the image, we preserve the background content and minimize distortion – empirically, we found editing the entire *source image* rather than just the *source person* produced lower quality edits with significant hallucination. For further implementation details, refer to the Appendix.

## 4.2 AUTOMATIC QUALITY FILTERING OF EDITED IMAGES

The synthetic edits with InstructPix2Pix Brooks et al. (2022) can often be of low quality or fail to edit the source person's attribute into the target attribute. In order to ensure the quality and gender accuracy of our synthetic image sets, we introduce an automatic filtering method using K-Nearest Neighbor (KNN), similar to Gu et al. (2020) who use KNN to score GAN-generated images.

First, we embed a collection of (i) source person bounding boxes, denoted as $R = \{r_1, r_2, ..., r_n\}$, and (ii) synthetically-edited person bounding boxes, denoted as $S = \{s_1, s_2, ..., s_m\}$ using CLIP. For each synthetic box $s_i$, we identify its K-nearest neighbors in this feature space, denoted as $N_{s_i} = \text{KNN}(s_i, R \cup S)$ using the Euclidean distance between the embeddings. If the proportion of real images within $N_{s_i}$, denoted as $P_R(s_i)$, and the proportion of images with the target gender of $s_i$, denoted as $P_G(s_i)$, exceed the thresholds $\tau_R$ and $\tau_G$ respectively, the edited image $s_i$ is accepted:

$$P_R(s_i) = \frac{1}{K} \sum_{r \in N_{s_i}} \mathbb{1}(r \in R) \quad \text{and} \quad P_G(s_i) = \frac{1}{K} \sum_{r \in N_{s_i}} \mathbb{1}(\text{gender}(r) = \text{gender}(s_i)), \quad (4)$$

$$\text{accept}(s_i) = \begin{cases} 1 & \text{if } P_R(s_i) > \tau_R \text{ and } P_G(s_i) > \tau_G \\ 0 & \text{otherwise.} \end{cases} \quad (5)$$

This process ensures that the accepted images are of high quality and accurately reflect the target gender change. We only retain images where the entire set of edits per unique COCO ID has at least one accepted male and female edit, then randomly select one edit for each gender from images that pass the filter. For examples of edits at each decile of $\tau_R$, see the Appendix.

## 4.3 VERIFYING THE QUALITY OF GENSYNTH

We evaluate the quality of the GENSYNTH dataset in three ways. First, we perform a human evaluation study where two annotators each assessed the perceived gender of 100 GENSYNTH images and 100 corresponding original COCO images. The results affirm our pipeline's successful gender-targeted editing, with high human agreement. Further details can be found in the Appendix. Second, we automatically measure the correctness of the targeted gender edit, by using CLIP to zero-shot classify the gender of people in the images. Third, to evaluate the semantic similarity of the edited image to the caption, we measure the text-to-image retrieval performance of CLIP on the synthetic text-image captions. For this, we edit the captions using the reverse procedure in Sec. 3.2 to reflect the gender of the person in the edited image. Then, for each image $I_i$ in GENSYNTH, where $i \in \{1, 2, ..., N\}$, we have a set of $n$ captions $C_i^j$, $j \in \{1, 2, ..., n\}$. For each caption $C_i^j$, we perform a retrieval operation from the COCO validation set combined with the query image $I_i$, to find a set of $K$ images that most closely match the caption, according to Euclidean distance of CLIP features. We denote this retrieved set as $R_i^j(K)$. The retrieval performance is evaluated using Recall at $K$ (R@K), which is defined as $R@K = \frac{1}{Nn} \sum_{i=1}^{N} \sum_{j=1}^{n} \mathbb{1}(I_i \in R_i^j(K))$.

We compare GENSYNTH, against (i) the original COCO 2017 dataset (train set) of natural images containing persons; and (ii) a weak gender-editing baseline – GENSWAP. This baseline has the same

Table 2: Dataset comparison between the original COCO dataset of natural person images and synthetically edited COCO from the GENSWAP and GENSYNTH pipelines. We report the presence of Spurious Background (BG) Correlations, Zero-Shot (ZS) Gender Accuracy, and Text-to-Image Retrieval Recall@K (R@K) amongst COCO Val 5k images using CLIP. *Unfilt.* refers to the synthetic pipeline without automatic quality filtering.

| COCO-Person Dataset | # Images | Edits per Image | Spurious BG. Correlations | ZS Gender Acc. (%) ↑ | Text-to-Image Retrieval ↑ | | |
|---|---|---|---|---|---|---|---|
| | | | | | R@1 | R@5 | R@10 |
| Original | 11,541 | - | ✓ | 93.6 | 30.9 | 54.4 | 64.9 |
| GENSWAP | 3,973 | 2 | ✗ | 67.9 | 19.0 | 39.8 | 50.4 |
| GENSYNTH (unfilt.) | 11,541 | 16 | ✗ | 83.9 | 22.4 | 43.4 | 53.8 |
| GENSYNTH | 3,973 | 2 | ✗ | 95.5 | 29.2 | 52.8 | 62.8 |

unique COCO images as in GENSYNTH, but only with edited faces – we replace the detected face in the COCO image with a random face of the target gender from the FairFace dataset Kärkkäinen & Joo (2021). Additional implementations of GENSWAP are provided in the Appendix.

As shown in Tab. 2, GENSYNTH leads to very similar zero-shot classification and retrieval results to the original COCO images. The filtering step significantly improves both metrics, successfully removing bad edits. The weak baseline, GENSWAP, consistently scores low, showing the importance of an effective editing method.

## 5 BENCHMARKING CLIP

### 5.1 EVALUATION SETUP

We use the following three datasets for evaluation: **GENSYNTH** consists of 7,946 images that have been generated and filtered as discussed in Sec. 4. It consists of 3,973 unique COCO images from the train set (62.6% of which were originally male), with a male and female edit for each. **COCO$_{\female \atop \male}$** consists of 3,973 original (unedited) images with the same unique COCO IDs as GENSYNTH. All images contain a single person, whose gender can be identified from the caption. **COCO$_{\female \atop \male \mathrm{Bal}}$** consists of 2,970 unique images from COCO$_{\female \atop \male}$, randomly sampled such that there is an equal number of male and female images. We use 5 different random seeds and report average results.

We compute MaxSkew@K for CLIP Radford et al. (2021) and CLIP-clip Wang et al. (2021a), with $m = 100$ clipped dimensions computed on COCO train 2017. We use the ViT-B/32 variant for both models. Refer to Appendix for evaluation with other debiased CLIP-like models. We only report MaxSkew@K, as we showed in Sec. 3.4 that Bias@K is not a suitable measure of model bias due to its signed nature – where for balanced data *with* spurious correlations it shows close to zero bias due to biases in both directions canceling each other out.

### 5.2 RESULTS

In Tab. 3 we compare the gender bias of the CLIP models for the three datasets defined in Sec. 5.1. We find debiased CLIP (CLIP-clip$_{m=100}$) records substantially lower bias on both unbalanced data (COCO$_{\female \atop \male}$) and balanced (COCO$_{\female \atop \male \mathrm{Bal}}$) data. This is because, as we noted in Sec. 3.4, balanced but biased data still contains spurious correlations related to gender. However, we observe that both models show very similar bias on the balanced and debiased [2] GENSYNTH data. This almost zero difference in bias is a result in itself – it means that the positive debiasing result on COCO$_{\female \atop \male \mathrm{Bal}}$ and COCO$_{\female \atop \male}$ are due to dataset bias. Overall, these findings suggest that intrinsic dataset bias, specifically spurious correlations, is artificially skewing the interpretations and comparisons of model bias. This reinforces the need for balanced data with no spurious correlations and shows the utility of our proposed pipeline for dataset debiasing.

---

[2]Balanced in respect to gender ratio and debiased in respect to spurious correlations.

Table 3: Comparison of Gender Bias between CLIP-like models on COCO-Person datasets. We report the MaxSkew@K in caption-to-image retrieval of gender-neutralised captions. We compare CLIP Radford et al. (2021) and CLIP-clip Wang et al. (2021a). We additionally report zero-shot image classification accuracy on ImageNet1K Deng et al. (2009).

| COCO-Person Dataset | Model | Gender Bias $\downarrow$ | | ImageNet1k Acc. (%) $\uparrow$ |
|---|---|---|---|---|
| | | MaxSkew@25 | MaxSkew@100 | |
| $COCO_{\female}^{\male}$ | CLIP | 0.27 | 0.20 | 63.2 |
| | CLIP-clip$_{m=100}$ | 0.23 | 0.16 | 60.1 |
| $COCO_{\female Bal}^{\male}$ | CLIP | 0.26 | 0.20 | 63.2 |
| | CLIP-clip$_{m=100}$ | 0.22 | 0.15 | 60.1 |
| GENSYNTH | CLIP | 0.23 | 0.18 | 63.2 |
| | CLIP-clip$_{m=100}$ | 0.22 | 0.17 | 60.1 |

## 6   LIMITATIONS AND ETHICAL CONSIDERATIONS

**Synthetic Shifts.** By generating synthetic data, we are creating a new evaluation distribution that does not necessarily represent the real-world distribution of the respective categories. This distribution shift can also be forced in contexts where it does not necessarily make sense to either face swap or make gender edits due to factual histories or biological identity Blodgett et al. (2021).

**Assumptions of Binary Gender.** Our data relies on the binary gender labels from the COCO and FairFace datasets and is necessarily influenced by our respective identities. COCO also presents limitations regarding race, ethnicity, and other sensitive attributes. We acknowledge this approach of using binary gender and making reference to perceived gender based on appearance oversimplifies the complexity of gender identity and biological sex, and risks erasing representation of non-binary people. Despite attempts to mitigate this limitation using terms such as "masculine" and "feminine", the resulting edits were often unusable (due to existing biases in generative models), necessitating reliance on binary and narrow terms. We advocate for future work that encodes and represents non-binary gender in datasets, and improves generalisation in generative models to non-binary terms.

**Stacking Biases.** Our pipeline may inadvertently introduce biases from the generative model via stereotypical representations of perceived gender, e.g., if "make this person more feminine" over-emphasises pink clothes, or "make this person more masculine" over-emphasises beards. The automatic filtering step also tends to favour images with simple scene arrangements. Some generated images were identified as NSFW, a consequence of training on large-scale internet datasets Birhane et al. (2021). Future work could integrate into our pipeline more capable and fair generative models.

## 7   CONCLUSION

The reliability of reported *model biases* in VLMs is affected by the interaction between *dataset bias* and choice of bias metric. In this paper, we demonstrated that naturalistic images from COCO have spurious correlations in image context with gender, which in turn affects how much trust can be placed in commonly-used metrics such as Bias@K: when measuring *model bias*, we may in fact be measuring *dataset bias*. To mitigate these problems, we proposed a pipeline for editing open-domain images at scale, creating gender-balanced contrast sets where the semantic content of the image remains the same except the person bounding box. Our method does not require manual auditing or image curation, relying instead on an effective automatic filtering method. Using this synthetically-created contrast set (GENSYNTH) we found that state-of-the-art CLIP-like models measure similarly on gender bias suggesting that measurements of model gender bias can largely be attributed to spurious model associations with gender (such as scene or background information) rather than gender itself. Through these subsequent angles of investigation, we conclude that only focusing on model bias while ignoring how dataset artefacts affect bias metrics paints an unreliable picture of identity-based bias in VLMs. We hope our work contributes to an ongoing discussion of how to seek improved representation and diversity of identity groups in image-captioning datasets, both now and in the future.

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

# Appendix

## A   IMPLEMENTATION DETAILS

Here we provide additional implementation details about our method.

### A.1   GENDERED WORDS AND CAPTION EDITING

In Tab. 4 we show the gendered words (Masculine, Feminine) that we use for assigning each caption a gender label. Captions without either a masculine or feminine word, or captions with matches from both of these lists are labeled as *undefined*. For switching or neutralising the gender in a caption, we map words across the rows of Tab. 4, so for example "she" could be replaced with "he" or "they". In Tab. 5 we show sentences that have been gender-neutralised.

Table 4: **Gendered word pairs.** We the Masculine and Feminine words in order to classify the gender of a person in an image given its caption. When editing the gender of a caption or making it gender-neutral, we use the word from the corresponding pair for the opposite gender or the gender-neutral word, respectively.

| Masculine | Feminine | Neutral |
|---|---|---|
| man | woman | person |
| men | women | people |
| male | female | person |
| boy | girl | child |
| boys | girls | children |
| gentleman | lady | person |
| father | mother | parent |
| husband | wife | partner |
| boyfriend | girlfriend | partner |
| brother | sister | sibling |
| son | daughter | child |
| he | she | they |
| his | hers | their |
| him | her | them |

Table 5: **Examples of gender-neutralised captions.** We show example original COCO captions with their gender-neutralised replacements, using the corresponding words from Tab. 4

| Original | Neutral |
|---|---|
| The **woman** brushes **her** teeth in the bathroom. | The **person** brushes **their** teeth in the bathroom. |
| A **man** sleeping with **his** cat next to **him**. | A **person** sleeping with **their** car next to **them**. |
| Two **women** and two **girls** in makeup and one is talking on a cellphone. | Two **people** and two **children** in makeup and one is talking on a cellphone. |

### A.2   IMAGE EDITING

Here we provide additional details on the two image editing pipelines in the paper – our proposed method GENSYNTH, and the weak baseline GENSWAP.

**GENSYNTH**   We edit the COCO train set images by applying InstructPix2Pix Brooks et al. (2022) on person crops (bounding boxes) with gender-editing instructions, as described in the main paper. We run InstructPix2Pix for 500 denoising steps, and for each instruction, we generate an image with two text guiding scales: 9.5 and 15. We found that a smaller guiding scale sometimes does not produce the required edit, whereas too large a scale results in an image that does not look natural. Using both scales ensures there are multiple candidates for the edited image, and then we can use the filtering pipeline to discard bad edits.

Table 6: **Discovered clusters in COCO Captions.** We show all 20 clusters with their manually assigned names, together with the top 10 words according to LDA. $\Delta$M represents the deviation from gender parity for males.

| Name | Words | $\Delta$M (%) |
|------|-------|---------------|
| dining$_{drinking}$ | wine, glass, holding, scissors, table, sitting, bottle, drinking, pouring, standing | -5.7 |
| dining$_{sweets}$ | cake, banana, donut, doughnut, holding, eating, candle, table, sitting, birthday | -14.0 |
| dining$_{mains}$ | pizza, eating, table, food, sandwich, sitting, holding, slice, hot, dog | -10.3 |
| sports$_{tennis}$ | tennis, court, racket, ball, player, racquet, hit, holding, swinging, playing | -6.0 |
| sports$_{snow}$ | ski, snow, slope, skiing, skier, snowboard, snowy, snowboarder, standing, hill | 4.7 |
| sports$_{skateboarding}$ | skateboard, skate, skateboarder, riding, trick, skateboarding, ramp, young, board, child | 27.9 |
| sports$_{ball}$ | baseball, bat, player, ball, soccer, field, pitch, holding, game, pitcher | 24.0 |
| sports$_{kite,frisbee}$ | frisbee, kite, playing, holding, field, beach, throwing, flying, standing, child | 11.6 |
| sports$_{surfing}$ | surfboard, wave, surf, surfer, riding, water, surfing, board, ocean, beach | 10.1 |
| sports$_{cycling,motorcycling}$ | motorcycle, riding, bike, bicycle, street, sitting, next, standing, ride, motor | 10.5 |
| leisure$_{street}$ | umbrella, holding, hydrant, standing, rain, fire, walking, street, child, black | -30.7 |
| leisure$_{park}$ | sitting, dog, bench, next, holding, park, child, two, sits, frisbee | -16.9 |
| formal attire | tie, wearing, suit, standing, shirt, glass, shirt, black, white, young | 19.7 |
| computer work | laptop, sitting, computer, bed, couch, desk, room, table, using, front | -4.6 |
| animals | horse, elephant, giraffe, riding, cow, standing, sheep, next, two, brown | -2.9 |
| video games | wii, game, remote, controller, playing, video, Nintendo, holding, room, standing | 4.8 |
| kitchen | kitchen, food, standing, refrigerator, oven, cooking, counter, chef, preparing, holding | -16.2 |
| bathroom | brushing, mirror, teeth, bathroom, cat, toothbrush, taking, toilet, holding, child | -14.0 |
| travelling | standing, bear, teddy, luggage, train, next, street, bus, holding, suitcase | -6.7 |
| phone calls | phone, cell, talking, holding, sitting, cellphone, standing, looking, wearing, young | -12.8 |

**GENSWAP** We use the MTCNN face detector Zhang et al. (2016) to detect faces in the COCO images (for the same subset in GENSYNTH), and replace them with faces from the FairFace repository Kärkkäinen & Joo (2021). FairFace is a collection of face crops from the YFCC-100M dataset Thomee et al. (2016), labeled with gender, race and age. We only use images whose age attribute is greater than 19 and randomly sample a face crop from the target gender.

### A.3 FILTERING

For the KNN filter, we set the neighbourhood size $K = 50$, and the thresholds $\tau_R = 0.08$ and $\tau_G = 0.5$.

## B SPURIOUS CORRELATIONS ANALYSIS

### B.1 USING DISCOVERED CLUSTERS VS COCO CLASSES

While prior works such as Plumb et al. (2021) use co-appearance of COCO classes, e.g. "tennis racket" and "person" to explore spurious correlations in COCO, we opt for discovering such keywords automatically from captions. We do so for two reasons. Firstly, using class co-occurrence simplifies the spurious correlations that exist in the dataset. For example, take the discovered clusters for leisure$_{street}$ and sports$_{cycling,motorcycling}$. Both appear on the street and considering co-occurence of COCO classes such as "car", "motorcycle", "water hydrant" could group the two clusters together. These two clusters exhibit opposite biases and if grouped together, would result in a close to zero overall bias. In contrast, captions refer to the activity the subject of the caption is performing, allowing us to separate semantically different activities. Secondly, our analysis only requires image captions, which are cheaper to obtain than object labels, and might be more generalizable to other datasets.

### B.2 DISCOVERED CLUSTERS

In Tab. 6 we show the 20 discovered clusters using K-Means, together with the top 10 salient words according to LDA. For each cluster, we show the male-overrepresentation factor, i.e., the difference between the percentage of images in that particular cluster relative to the percentage of male images in the person class of COCO as a whole.

## C    PROMPT EDITING TEMPLATES

Tab. 7 contains the complete set of edit instructions input to InstructPix2Pix to edit the single person bounding box for each attribute label.

Table 7: Templates used for prompt editing.

| Template | Instruction | |
| --- | --- | --- |
| | **Feminine** | **Masculine** |
| Make this person more { } | feminine | masculine |
| Make this person look like a { } | woman | man |
| Turn this person into a { } | woman | man |
| Convert this into a { } | woman | man |

## D    HUMAN EVALUATION STUDY

Each of two annotators annotated the perceived gender of 100 images from the GENSYNTH dataset. They then annotated the perceived gender of the 100 original COCO images corresponding to the same IDs. The 100 GENSYNTH images were randomly sampled from the dataset without replacement so there were no repeats and no overlap between annotators. For the first annotator, their given labels matched the GENSYNTH gender label in 99% of images (99 images), and their given label matched the COCO original gender label in 95% of images. For the second annotator, there was 95% agreement in gender labels for the GENSYNTH images and 98% agreement in the COCO original images. In sum, these results show that our pipeline successfully edits the subject of the image to the target gender (e.g., from a man to a woman) as demonstrated by the high levels of human agreement.

## E    EXTENDED BENCHMARKING OF CLIP

Here we extend the analysis of CLIP models in the main paper. We evaluate the following models: (i) CLIP Radford et al. (2021); (ii) CLIP-clip Wang et al. (2021a), with $m = 100$ clipped dimensions computed on COCO train 2017; (iii) DebiasCLIP Berg et al. (2022), which has been debiased on the FairFace dataset; and (iv) OpenCLIP Ilharco et al. (2021) models trained on LAOIN 400M and 2BN datasets Schuhmann et al. (2022). We use the ViT-B/32 variant for all models, except for DebiasCLIP, for which ViT-B/16 is used due to its availability from the authors.

In Tab. 8 we make a similar observation to the one discussed in the paper, where debiased CLIP models perform on par with other CLIP models on GENSYNTH.

## F    ABLATION STUDY

We ablate the use of a CLIP vision encoder in the KNN filtering pipeline. We replace it with a DINO ViT-B/16 Caron et al. (2021) and repeat the analysis. We found that using DINO features is much more powerful when it comes to discriminating between the different images (real versus fake), and that the male and female images are better clustered. Accordingly, for the real vs. fake filter we use a neighborhood size of $K = 5,000$ and a threshold $\tau_R = 0.0002$ (i.e., the generated images have at least *one* real neighbour). For the male vs. female filter, we use a neighborhood size of $K = 50$ and a threshold $\tau_G = 0.4$. We end up with 571 unique COCO images, or 1,142 images in total (with a male and female edit for each unique image). The R@K results with this dataset are R@1 = 33.7%, R@5 = 57.1% and R@10 = 66.7%, and the zero-shot gender classification accuracy is 87.4%. Due to the different filtering, this dataset (with DINO filtering) is smaller than GENSYNTH and the results have higher variance, but are comparable to GENSYNTH.

We evaluate MaxSkew@K on this dataset in Tab. 9. We observe a similar trend to the GENSYNTH dataset, where bias results across models have a smaller variance than results on the unbalanced and balanced COCO$_{\bar{x}}$ datasets. The absolute values of the bias metric are smaller, which we explain with the different images retrieved, and the variance that comes with that.

Table 8: Comparison of Gender Bias between CLIP-like models on COCO-Person datasets. We report the MaxSkew@K in caption-to-image retrieval of gender-neutralised captions. We compare CLIP Radford et al. (2021) and CLIP-clip Wang et al. (2021a), DebiasCLIP Berg et al. (2022), and OpenCLIP Ilharco et al. (2021) trained on LAOIN 400M & 2BN Schuhmann et al. (2022). We additionally report zero-shot image classification accuracy on ImageNet1K Deng et al. (2009).

| COCO-Person Dataset | Model | Gender Bias ↓ | | ImageNet1k Acc. (%) ↑ |
|---|---|---|---|---|
| | | MaxSkew@25 | MaxSkew@100 | |
| COCO$_{\female\male}$ | CLIP | 0.27 | 0.20 | 63.2 |
| | CLIP-clip$_{m=100}$ | 0.23 | 0.16 | 60.1 |
| | DebiasCLIP | 0.29 | 0.22 | 67.6 |
| | OpenCLIP$_{400M}$ | 0.26 | 0.20 | 62.9 |
| | OpenCLIP$_{2B}$ | 0.27 | 0.21 | 65.6 |
| COCO$_{\female\male Bal}$ | CLIP | $0.26_{\pm 0.00}$ | $0.20_{\pm 0.00}$ | 63.2 |
| | CLIP-clip$_{m=100}$ | $0.22_{\pm 0.00}$ | $0.15_{\pm 0.00}$ | 60.1 |
| | DebiasCLIP | $0.28_{\pm 0.01}$ | $0.21_{\pm 0.00}$ | 67.6 |
| | OpenCLIP$_{400M}$ | $0.27_{\pm 0.00}$ | $0.20_{\pm 0.00}$ | 62.9 |
| | OpenCLIP$_{2B}$ | $0.27_{\pm 0.00}$ | $0.21_{\pm 0.00}$ | 65.6 |
| GENSYNTH | CLIP | 0.23 | 0.18 | 63.2 |
| | CLIP-clip$_{m=100}$ | 0.22 | 0.17 | 60.1 |
| | DebiasCLIP | 0.24 | 0.19 | 67.6 |
| | OpenCLIP$_{400M}$ | 0.24 | 0.19 | 62.9 |
| | OpenCLIP$_{2B}$ | 0.23 | 0.18 | 65.6 |

Table 9: Comparison of Gender Bias between CLIP-like models on the accepted images using DINO image embeddings for KNN filtering. We report the MaxSkew@K in caption-to-image retrieval of gender-neutralised captions. We compare CLIP Radford et al. (2021), CLIP-clip Wang et al. (2021a). We additionally report zero-shot image classification accuracy on ImageNet1K Deng et al. (2009).

| COCO-Person Dataset | Model | Gender Bias ↓ | | ImageNet1k Acc. (%) ↑ |
|---|---|---|---|---|
| | | MaxSkew@25 | MaxSkew@100 | |
| GENSYNTH (DINO) | CLIP | 0.15 | 0.12 | 63.2 |
| | CLIP-clip$_{m=100}$ | 0.13 | 0.10 | 60.1 |

# G  QUALITATIVE DATASET EXAMPLES

In Fig. 3, we show gender edits for the GENSYNTH and GENSWAP datasets, alongside the original COCO image and ID. The GENSYNTH edits are more naturalistic than the GENSWAP edits, and also make changes to the body or clothing of the subject.

# H  COMPARING IMAGE EDITS ACROSS FILTERING THRESHOLDS

For each edited image, we calculate $P_R$, i.e., the ratio of real images versus fake images in the KNN clustering step. We then average $P_R$ for each *pair* of images (the male and female edit). In Fig. 4, we show these randomly-selected pairs of gender edits from each decile of averaged $P_R$ to demonstrate how our threshold filtering step improves the quality of the edited images.

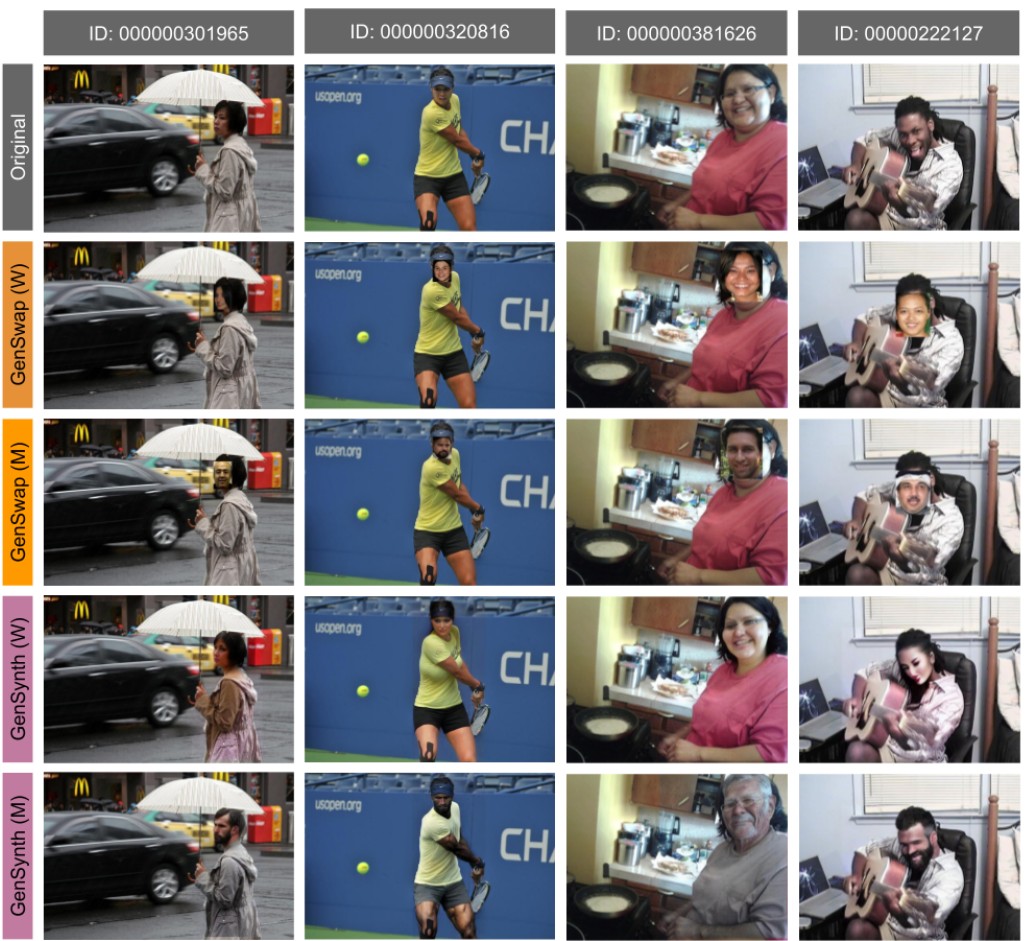

Figure 3: Randomly selected examples of GENSYNTH images showing a comparison to the original COCO image and the weak baseline GENSWAP.

Figure 4: Averaged KNN Score ($P_R$) for pairs of edited images using the GENSYNTH pipeline.

1st to 4th decile of scores.

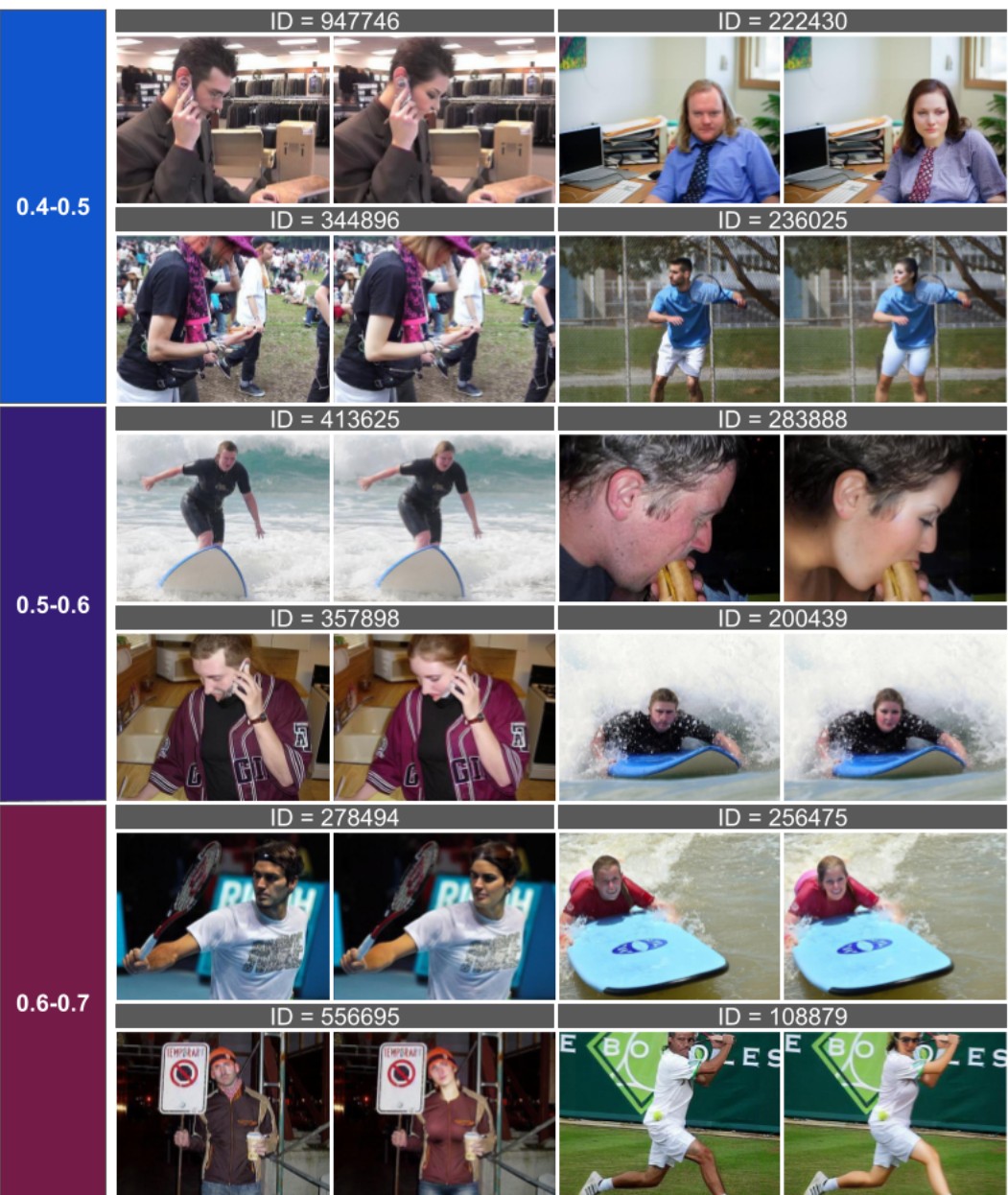

5th to 8th decile of scores. Note that there was only one image with an averaged score between 0.7-0.8, and no images in the higher deciles.

