# OpenReview forum: "Balancing the Picture: Debiasing Vision-Language Datasets with Synthetic Contrast Sets"
_ICLR.cc/2024/Conference — Submitted to ICLR 2024_

### Official Review · Reviewer_YKtn · 2023-10-25

**Soundness:** 3 good
**Presentation:** 3 good
**Contribution:** 2 fair
**Rating:** 3
**Confidence:** 4

**Summary:**

This work aims to address and understand biases in vision-language datasets, specifically the bias between the background context and the gender of persons referenced or appearing in the data. Demonstrating their approach on the COCO Captions dataset with CLIP-based models, the authors propose a method for modifying the dataset with synthetically generated “contrast sets” where the gender of the subject is edited using InstructPix2Pix and the background fixed, such that the dataset can be balanced in gender.

**Strengths:**

* This work tackles the important problem of understanding and addressing dataset bias.
* The paper is relatively easy to understand.
* To the best of my knowledge, this appears to be a novel approach for editing an attribute (gender) in an image and observing how it impacts dataset bias.

**Weaknesses:**

* Looking at qualitative examples of the GenSynth and GenSwap gender-edited images (Fig 1 in the appendix), the resulting images still have obvious artifacts, even after the various steps taken to verify quality and filter out low-quality images. From what I can tell, the authors have not really addressed how these artifacts may be a factor in their results.
* I would be interested to see a comparison with other image editing methods other than InstructPix2Pix that may produce higher quality contrast sets. How does varying the quality of the edited images affect the observed gender biases for the different models?
* The work only debiases for a single attribute (gender), and does not delve into what debiasing multiple attributes may look like. This limits the potential impact of this work (it’s not quite usable in practice, and the observations about bias in CLIP are also limited to gender only).
* The instructions for InstructPix2Pix only use gender editing, and does not control for other variables (e.g. skin tone, race).

**Questions:**

* The Wang et al. 2021a paper is referenced frequently; it would be useful to the reader to discuss in more depth how this work differs from the former.
* There is limited discussion of the CLIP models used; given how prominently they feature in the experiments, I would expect the authors to provide more background for the reader.
* In general, I found that the Appendix contained a lot of relevant information (e.g. qualitative examples) that I would have liked to see in the main paper.
* Section 3.2 “Extracting Image Gender Labels...”: The authors mention that the “images may be incorrectly labeled as undefined”. What percentage of the images were mislabeled?

---

> ### Author Response · Authors · 2023-11-21
> **Response to Reviewer YKtn**
>
> We thank the reviewer for the insightful comments and questions, and we provide the responses below.
>
> # Weakness 1
> We agree with the reviewer that even after applying our filtering step artefacts in the images remain. We preferred having an automatic filtering method instead of manually auditing and curating our dataset, as we believe this would make it easier for future researchers to use our proposed pipeline off the shelf. Despite having artefacts, the semantic content of the images remained the same after applying our method and we believe, through our spurious correlations investigations, that this is in fact the biggest predictor of gender in an image, rather than the gender of the person in the image, i.e. the background is associated with a certain gender and the pre-trained models learn this instead of focusing on the actual gender of the person in the image. Therefore, even if some artefacts in our images remain, we show that there is sufficient evidence to believe measurements of model gender bias can largely be attributed to spurious model associations with gender rather than gender itself.
>
> # Weakness 2
> While performing our experiments, we experimented with other open-source image-editing generative models, namely null-text inversion (https://null-text-inversion.github.io/). We found no differences between this editing method relative to InstructPix2ix. We opted to use InstructPix2Pix for our experiments due to its novelty and very good image editing capabilities. We believe that our pipeline will greatly benefit from the rapid advancements in the field of instruction-based image generative models since more accurate gender edits will be possible, making our evaluations of model bias more accurate. However, we also point the reviewer to the answer to Weakness 1 and note that it may be the case that gender bias measurements are not significantly tied to the quality of edited images, since we found the background of an image to be the best predictor of gender in an image.
>
> # Weakness 3
> We understand the reviewer’s concern that other biases could be analysed in this work. We refer the reviewer to our answer to Weakness 1 of Reviewer Nd3Y.  We would like to add that our pipeline is readily applicable to other scenarios, as long as the datasets contain captions with rich annotations that allow the extraction of relevant information regarding relevant attributes.
>
> # Weakness 4
> We decided to focus only on gender bias as it was impossible to extract information regarding skin tone, race etc from the COCO captions, making it impossible to provide InstructPix2ix with instructions to control for such attributes. As a result, we kept our gender editing instructions as generic as possible to try to edit the gender of the person in an image as best as possible. Our instructions are limited to what information is possible to extract from the captions of the dataset currently being utilised, and we acknowledge this as a limitation of our pipeline.
>
> # Question 1
> Wang et al. 2021a take a pre-trained model (CLIP) and find the dimensions of image feature embeddings that are highly correlated with gender information and clip the $m$ features that have the highest mutual information with gender attributes. They argue that this procedure removes the image features that are highly correlated with gender attributes. In our work, we show that the method proposed by Wang et a. 2021a (CLIP-clip) (i) records a substantially lower bias on the original COCO evaluation set than non-debiased models BUT (ii) records very similar bias to other models (whether debiased or not) on the balanced and unbiased GenSynth dataset (our proposed method). These findings in tandem suggest that intrinsic dataset bias is artificially skewing our interpretations and comparisons of model bias. The need for a balanced unbiased dataset to reliable measure model bias motivated us to propose GenSynth. Our work differs from the work of Wang et al. 2021a in that we do not propose an “unbiased” model, instead we propose a synthetically-created contrast set dataset (GenSynth) that allows reliably measuring model bias.
>
> # Question 2
> We focussed on CLIP as this is a well-known VLM. However, we acknowledge that the background we provide on CLIP models may be insufficient for the reader and we will rectify this issue in the revised manuscript.
>
> # Question 3
> We agree with the reviewer that the Appendix contains relevant information for the reader. In our revised manuscript we will aim to better refer the reader to the relevant Appendix sections.
>
> # Question 4
> There are 75,965 (64.2%) undefined images in the COCO 2017 train set. We did not include images which had zero or more gender pronouns. We can report that of those 75,965 undefined images, 74,271 had no gender pronouns and 1,694 had multiple gender pronouns. For the COCO 2017 validation set there were 3,186 undefined images, 3,118 had no gender pronouns and 68 had multiple gender pronouns.

---

### Official Review · Reviewer_E6KN · 2023-10-31

**Soundness:** 3 good
**Presentation:** 3 good
**Contribution:** 2 fair
**Rating:** 5
**Confidence:** 4

**Summary:**

This paper introduces a novel gender-debiased dataset named GENSYNTH. The dataset aims for gender balance and context independence, achieved through the use of image generation models guided by prompts, which allow for control over the gender expression of the generated images, ranging from masculine to feminine. The experimental results, as depicted in Table 3, demonstrate that the utilization of the GENSYNTH dataset effectively reduces gender bias.

**Strengths:**

+ Addressing this issue is crucial for the machine learning research community.
+ The methodology employed presents a plausible solution for mitigating gender bias.
+ Utilizing generative models as a strategy for reducing bias holds significant promise.

**Weaknesses:**

I recognize that this paper addresses a critical issue; however, I believe it is not yet suitable for publication due to numerous absent discussions.

- Technical novelty is weak:
While the methodology is intriguing and the results are persuasive, the strategy appears somewhat simplistic, essentially constituting an application of generative models.

- More detailed discussion should be conducted:
The instance depicted in Fig. 2 is gender-neutral. Nonetheless, in certain scenarios, other contexts might exhibit stronger gender biases, such as facial hair, attire, etc. Thus, merely manipulating images to appear more masculine or feminine does not consistently resolve the issue. The authors should explore and discuss the potential repercussions of generating new "noisy" or "misleading" examples.

- More detailed discussion on bias freeness is needed:
The experimental results section predominantly presents numerical data, with a relatively modest improvement in performance compared to the baseline. The authors are encouraged to illustrate typical instances where gender bias mitigation is evident, as well as highlight persisting challenges, acknowledging that complete eradication of gender bias is unfeasible with the proposed method.

- The potential introduction of new biases requires careful consideration:
The generative models are already acknowledged to be biased. The authors should thoroughly investigate potential risks associated with utilizing generative models for debiasing objectives. For example, in Fig. 3, adjustments are made to not only facial features but also skin color and clothing color, potentially leading to the inception of new biases. Furthermore, it is observed that the generative models seem to underrepresent Asian faces, potentially introducing additional biases.

**Questions:**

Please answer to my “weakness part.”

---

> ### Author Response · Authors · 2023-11-21
> **Response to Reviewer E6KN**
>
> We thank the reviewer for the insightful comments and questions, and we provide the responses below.
>
> # Weakness 1
> We thank the reviewer for the valuable comment. We agree with the reviewer that the bulk of our pipeline consists in using a generative model to edit the gender of an image while keeping the background the same. However, despite our method being “straightforward”, we believe we show clear evidence that previous findings assessing model bias are due to spurious correlations in the data used for evaluation and thus not reflective of actual model bias. We seeked to show that dataset artefacts affect bias metrics and paint an unreliable picture of identity-based bias in VLMs. To mitigate this and allow a fairer and more robust comparison of model bias we propose our pipeline for editing images which does not require any manual supervision and provides researchers a reliable and easy way to assess bias in their models. We believe our novelty resides mostly in showing how model bias measurements are unreliable and proposing a clever way using generative models to mitigate this issue.
>
> # Weakness 2
> We thank the reviewer for the comment. We believe the images produced using the pipeline we propose look natural and realistic. The way we set up the KNN filtering step ensured the filtered images were similar in embedding space to the real images, thus ensuring that they looked as much as possible to real images while changing the gender attribute. Furthermore, when performing the human study in Appendix D we observed that the images looked natural. However, not all images were free of artefacts and we realise that our method may have failure cases when dealing with stronger gender biases as the reviewer suggests. Although our method may generate noisy examples, we believe the way we set up our entire pipeline mostly avoids this and we also believe our pipeline is flexible enough to integrate new developments in instruction-based generative editing models which would decrease the risk of generating such noisy instances.
>
> # Weakness 3
> We thank the reviewer for the valuable comment. We acknowledge that complete eradication of gender bias is unfeasible with the proposed method or, possibly, with other proposed methods that may exist since as long as our datasets contain certain biases these will always be learned and perpetuated by our generative models. However, having said that, we do believe our proposed method does a very good job at editing the gender attribute of a person in the image as shown in our examples in Figure 3. To add to this we show that measurements of model gender bias can largely be attributed to spurious model associations with gender (such as scene or background information) rather than gender itself, meaning that VLMs may effectively be learning more about the background of images and their associations with certain attributes in a person rather than focusing on the person itself which would mean failure cases of our method would not be extremely detrimental to the point we try to convey in our work. However, we do recognise that challenges remain, for instance we focus on binary gender, we oversimplify the complexity of gender identity and biological sex and we are limited by the capabilities of current image editing generative models. The last point specifically is extremely relevant to our work, as the success of our pipeline is intrinsically connected to how capable and reliable these image generative models are at editing specific attributes of an image.
>
> # Weakness 4
> We thank the reviewer for the insightful comment. We would like to point the reviewer to our answer to Weakness 2. Furthermore, we would like to add that this was indeed one of our concerns, introducing additional biases due to the biases present in the generative model that was utilised. This is something we acknowledge is possible with our pipeline, but it was impossible for us to fully control. As such, to the best of our ability, we introduced the KNN filtering step to ensure the newly generated images were as similar as possible to the original image while changing the gender. Finally, we are aware that Asian faces are underrepresented, as are other races, ethnicities, gender entities and other relevant attributes, and that is precisely one of the reasons that motivated this work, to contribute to an ongoing discussion on how to seek improved representation and diversity of identity groups in image-captioning datasets. We hope our work shows that claims for unbiased models are misleading and this is mostly due to the fact that current datasets do not encompass a lot of the variability regarding different attributes we see in our world.

---

### Official Review · Reviewer_Nd3Y · 2023-11-05

**Soundness:** 3 good
**Presentation:** 3 good
**Contribution:** 4 excellent
**Rating:** 6
**Confidence:** 3

**Summary:**

In this work, author target the problem of data debiasing, and propose a pipeline to augment the COCO to generate an synthetic, gender-balanced contrast sets, by editing the person in the image without shifting the background, to prevent spurious relationship between gender and background. By using the generated dataset, authors shows that the conventional metric for measuring model bias, are highly biased by the bias from dataset. Author appeal for attention on dataset debiasing to the community.

**Strengths:**

1. Author identify an vital problem in existing metric of measuring model debiasing, is less accurate due to being skewed by dataset bias. This provide new insight to the community and could be potential impactful.
2. Author also identify background bias as a vital source for gender bias by showing result from spurious correlations classifier.
3. Author provide viable framework to generate balanced dataset without spurious relationship, and propose a dataset under this framework.
4. Most discussion of the paper is clear and easy to follow.

**Weaknesses:**

1. In this work, author only adopted COCO Captions dataset for tasking and gender as the debias attribute. It would be more convincing if author provide discussion and empirical result of how does insight draw from this work also applicable to other dataset and attribute.
2. In section 3 and 5, author shows that standard metric are biased by spurious relationship in dataset itself. However, there's no followup discussion on how to measure model bias over constructed balance, spurious-relationship free dataset.

**Questions:**

The insight that spurious relationship between background pixel and foreground object seems to be generalizable beyond VLM and caption based dataset.  It would be helpful if authors can also provide some discussion on that. Also how does this insight be potential beneficial to other field like bias-free image generation?

---

> ### Author Response · Authors · 2023-11-21
> **Response to Reviewer Nd3Y**
>
> We thank the reviewer for the insightful comments and questions, and we provide the responses below.
>
> # Weakness 1
> > In this work, author only adopted COCO Captions dataset for tasking and gender as the debias attribute. It would be more convincing if author provide discussion and empirical result of how does insight draw from this work also applicable to other dataset and attribute.
>
> We thank the reviewer for the valuable comment. We understand the reviewer’s concern that other datasets and attributes could be analysed in this work. However, our method requires using a dataset that has sufficient attribute information in either the caption or meta-data for each image. This motivated the use of the COCO Captions dataset given its high quality image-caption pairs, from which gender labels are easily mined. Other attributes, such as age, are less commonly described in captions and may have a lower inter-annotator agreement, so we decided to only analyse the gender attribute.
>
> # Weakness 2
> > In section 3 and 5, author shows that standard metric are biased by spurious relationship in dataset itself. However, there's no followup discussion on how to measure model bias over constructed balance, spurious-relationship free dataset.
>
> We thank the reviewer for the valuable comment. In this work, we are concerned with showing that, in order to properly measure gender bias, datasets should be balanced and free of spurious gender correlations. We claim that the way gender bias is being measured is inadequate since datasets are often unbalanced and contain spurious correlations and, as a result, standard gender bias metrics won’t provide a fair representation of gender bias for a given pre-trained model. However, we do not attempt to propose new gender bias metrics, as that is beyond the scope of this work. We simply show that, given the current bias metrics available in the field, balanced and spurious-relationship free datasets should be used to measure gender bias. To conclude, we do hope that with this paper researchers focus their attention on proposing better and more robust bias metrics.
>
> # Question
> > The insight that spurious relationship between background pixel and foreground object seems to be generalizable beyond VLM and caption based dataset. It would be helpful if authors can also provide some discussion on that. Also how does this insight be potential beneficial to other field like bias-free image generation?
>
> We thank the reviewer for the insightful question and the opportunity to discuss this further. We agree with the reviewer that spurious correlations between the background and foreground are generalizable beyond VLM and caption-based datasets, in fact it is something ubiquitous about our world and across multiple attributes, such as age, race, skin colour, ethnicity. We believe that such correlations are part of our world and will always be present in our datasets. The problem is when such spurious correlations are harmful or perpetuate and amplify social stigmas and prejudices we have present in our society. When this happens, we are in fact training our models (LLMs, VLMs, etc) to learn such societal biases which can then have detrimental and harmful effects in downstream applications. However, we do recognise that having a completely balanced and unbiased dataset is perhaps an impossible endeavour. This is why we were motivated to show that measurements of model gender bias can largely be attributed to spurious model associations with gender (such as scene or background information) rather than gender itself, and this is true for other attributes. We recognise that there will always be biases present in the data, but what we think is detrimental to the field and society is claiming that a given model, which can for instance be deployed in a real-world setting, is unbiased when, in fact, it is not. In regards to the field of bias-free generation, we believe the same insights apply. We believe we need to recognise that our current image generative models are biased because they are trained on uncurated image-text pairs from the internet. However, it is not straightforward how to make such image generation processes bias-free, as curating entire internet datasets is an infeasible task. The easiest way towards building bias-free image generation methods would be to incorporate some of this knowledge regarding spurious correlations into the model, but this is a non-trivial task that we did not attempt to address in this work, but that we hope we have motivated more researchers to pursue.

---

> > ### Comment · Reviewer_Nd3Y · 2023-11-23
> >
> > Thanks reviewer for the detail reply. I keep my rating score of 6.

---

### Meta-Review · Area_Chair_Wqf3 · 2023-12-12

**Metareview:**

The authors propose to use a generative model to debias multimodal MS-COCO dataset, which contains spurious correlation between the background and the gender of the person in the image. While the proposed method / problem are interesting, the reviewers pointed out some subtle issues that need to be addressed carefully before a publication of the work. For example, some fine-grained attributes such as facial hair / attire could be spuriously correlated with the gender as well, and a simple conversion by instruct Pix2Pix may not guarantee unbiased gender conversion. (As EK6N pointed out.) But, we believe the work has a potential, so incorporating the constructive comments from the reviewers may strengthen the paper for a future submission.

**Justification For Why Not Higher Score:**

As mentioned above, some subtle points regarding the spurious correlation as well as considering other correlations could be necessary for strengthening the paper.

**Justification For Why Not Lower Score:**

N/A

---

### Decision · Program_Chairs · 2024-01-16

Reject